# Glucosylation of Isoeugenol and Monoterpenes in *Corynebacterium glutamicum* by YdhE from *Bacillus lichenformis*

**DOI:** 10.3390/molecules28093789

**Published:** 2023-04-28

**Authors:** Su Yeong Ma, Obed Jackson Amoah, Hue Thi Nguyen, Jae Kyung Sohng

**Affiliations:** 1Department of Life Science and Biochemical Engineering, Sun Moon University, 70 Sun Moon-ro 221, Tangjeong-myeon, Asan-si 31460, Republic of Korea; su970130@gmail.com (S.Y.M.); jacksonamoahobed@gmail.com (O.J.A.); huenguyencute@gmail.com (H.T.N.); 2Department of Pharmaceutical Engineering and Biotechnology, Sun Moon University, 70 Sun Moon-ro 221, Tangjeong-myeon, Asan-si 31460, Republic of Korea

**Keywords:** *O*-Glucosylation, *Corynebacterium glutamicum*, monoterpene, acetylated glucoside

## Abstract

*Corynebacterium glutamicum* has been regarded as a food-grade microorganism. In recent years, the research to improve the activities of beneficial therapeutics and pharmaceutical substances has resulted in the engineering of the therapeutically favorable cell factory system of *C. glutamicum.* In this study, we successfully glucosylated isoeugenol and other monoterpene derivatives in *C. glutamicum* using a promiscuous YdhE, which is a glycosyltransferase from *Bacillus lichenformis*. For efficient glucosylation, cultivation conditions such as the production time, substrate concentration, carbon source, and culture medium were optimized. Our system successfully converted about 93% of the isoeugenol to glucosylated compounds in the culture. The glucoside compounds were then purified, analyzed, and identified as isoeugenol-1-*O*-β-d-glucoside and isoeugenol-1-*O*-β-d-(2″-acetyl)-glucoside.

## 1. Introduction

*Corynebacterium glutamicum* is a Gram-positive, facultative anaerobic, rod-shaped, non-pathogenic, non-spore-forming bacteria that is generally regarded as safe (GRAS) because it does not produce endotoxin [1]. *C. glutamicum,* previously named *Micrococcus glutamicus* after its 1957 discovery in Japan, produced a massive amount of glutamate, thus revolutionizing the world of biotechnology [2]. Since then, *C. glutamicum* has been extensively employed in the food industry for the synthesis of various amino acids, such as L-glutamate, L-lysine, and L-arginine [3,4,5,6,7]. It is primarily used for large-scale monosodium glutamate (MSG) biosynthesis due to its characteristics as a fermentative bacterium [8]. It has also been reported to be a promising general-purpose chassis strain for additional useful chemicals [9,10,11], and a cutting-edge host for the production of heterologous proteins [7,12]. Recently, *C. glutamicum* ATCC 13032 has been successfully employed for the biosynthesis of C_40_ and C_50_ carotenoids as well as glucosylated compounds such as UDP-N-acetylglucosamine, which are difficult to synthesize [13,14], and the glucosylation of various *N*-linked therapeutic and bioactive compounds to enhance their poor solubility [15]. In addition, the flexibility of the cell factory of *C. glutamicum* and its wide substrate spectrum and high production yield have led to the metabolic engineering of its system for the biosynthesis of various organic compounds. For example, metabolically engineered *Corynebacterium glutamicum* achieved a comparatively higher production yield of 8.6 g/L than 3-hydroxycadevarine [16].

Essential oils are secondary metabolites produced by aromatic plants such as plants, herbs, or spices. They are volatile, natural, complex compounds with a strong odor [17,18]. Additionally, depending on their chemical structure, metabolic degradation products from plants are classified into various groups: terpenes, terpenoids, phenylpropenes, and other constituents [19]. Due to their antioxidant, antibacterial, and aromatic flavoring effects, they have generated a great deal of attention as a natural source of food additives [20,21]. In addition, they can be applied to the perfume and cosmetics industries [22]. Despite having numerous beneficial functions, they are highly water-insoluble, which limits their use in the biotechnology industry [23]. The main challenge in converting these compounds into other derivatives is related to the loss of some pharmacological properties and their natural flavors [24]. As a result, studies trying to solve the solubility problem of some monoterpenes, such as eugenol and isoeugenol, have been actively pursued, thus far by using glycosylation, which is the attachment of sugar moieties to aglycones, thereby enhancing their water solubility while retaining their inherent flavor and stability. Furthermore, several biological functions and properties are known to be displayed by sugar-conjugated natural compounds [25,26,27,28,29,30,31,32].

Isoeugenol, a fragrance chemical with a spicy, carnation-like perfume and a well-known mild human sensitizer, is a member of the phenylpropene group of essential oils, one of the four groups of essential oil [33]. However, some factors, such as low water solubility and high volatility, contributed to the marked aromatic note, limiting its applications in the food and feed industries [34]. Water solubility is one of the most significant physicochemical parameters used in the cosmetics industry to assess skin absorption [35]. Hence, approaches are being developed to improve the solubility of weakly soluble compounds, such as isoeugenol complexed with HP*β*CD [36], and eugenol nanoliposomes [37]. The chemical method, however, has limitations, including strict reaction conditions and being time-consuming when employed to improve weak solubility in water. In addition to the use of chemical approaches for the glucosylation of therapeutic compounds, microorganisms, particularly *Escherichia coli,* have been widely employed and metabolically engineered for the glucosylation of both natural and non-natural products. However, the use of an *E. coli* cell factory system is limited by the production of endotoxins, low substrate tolerance, and low yield of products [38,39]. Hence, there is a need to employ the cell factory of the safe *C. glutamicum*, which also possesses the advantage of having a higher substrate tolerance for the large-scale biosynthesis of therapeutic products. In this study, we described the development and the methods for the enhanced and effective glucosylation of isoeugenol, an isomer of eugenol, in higher quantities with an in vivo system. We expressed YdhE, a highly promiscuous enzyme from Bacillus lichenformis, which is known to take a wide range of structurally varied compounds as a substrate, in engineered *C. glutamicum* [15]. YdhE was able to attach glucose moieties to isoeugenol to produce isoeugenol glucoside. Interestingly, it was observed that the constitutive sugar *O*-acetyltransferase gene (Protein ID; AUH99773.1), attached the acetyl group to the glucoside compound, yielding both glucoside and acetyl glucoside compounds (Figure 1).

## 2. Results and Discussion

### 2.1. Biotransformation of Isoeugenol in C. glutamicum Expressing CO-YdhE

The crude extract with isoeugenol from *C. glutamicum* harboring the recombinant plasmid pSKSM-YdhE showed two peaks in the HPLC results. Two new peaks of **1b** and **1c** were detected at a t_R_ of 10.83 min and 11.73 min, respectively, with UV absorbance at 254 nm (Figure 2). Two samples were then analyzed by high-resolution quadrupole time-of-flight electrospray ionization mass spectrometry (HR-QTOF ESI/MS). The compound mass fragment of **1b** [M + Na]^+^
*m*/*z* = 349.1261 was matched to the calculated mass [M + Na]^+^
*m*/*z* = 349.1258 (C_16_H_22_NaO_7_^+^). Likewise, the compound mass fragment of **1c** [M + Na]^+^
*m*/*z* = 391.1354 was matched to the calculated mass [M + Na]^+^
*m*/*z* = 391.1363 (C_18_H_24_NaO_8_^+^) in the positive ion mode (Figure 3), which resembled the glucosylated and glucose-acetylated derivatives of isoeugenol.

### 2.2. Structure Determination of Two Isoeugenol Derivatives ***1b*** and ***1c***

The analytical HPLC metabolite peak eluates were collected, concentrated under a vacuum, and then lyophilized to yield the purified compounds **1b** (17 mg) and **1c** (13 mg) from the crude extracts. Nuclear magnetic resonance (NMR) was used to determine the chemical structures of compound **1c** and identified **1b**, which was reported by Bashyal et al. 2019 [27]. The ^1^H NMR spectra of **1b** and **1c** both showed the presence of a proton anomeric signal at signal *δ* 3.32–3.22 (*m*) and *δ* 4.91 (*d*). The signal at 1.99 ppm (*s*) confirmed the attachment of one acetyl group of a glucose molecule to the aglycone with a β-configuration, which is different from the ^1^H NMR spectrum of **1b** (Appendix A). In the ^13^C NMR spectrum, the signals at *δ* 21.12 ppm and *δ* 170.70 ppm were assigned to the carbons of one acetyl group, which was different from the ^13^C NMR spectrum of compound **1b** (Appendix A). Eighteen carbons were obtained from the ^13^C NMR spectrum of compound **1c**, confirming the molecular formula of C_18_H_24_O_7_. Furthermore, to confirm the correlation of aglycon protons and glucose moiety protons, two-dimensional (2D)-NMR analysis and ^1^H–^1^H COSY was performed. According to ^1^H, ^13^C NMR, and COSY NMR spectra results, **1c** was confirmed as isoeugenol-1-*O*-β-d-(2″-acetyl)-glucoside (Table 1, Appendix A).

### 2.3. Production of Isoeugenol Glucoside

To enhance the synthesis of the glucoside derivatives, the culture conditions were optimized using isoeugenol that was substrate-fed into *C. glutamicum*. The optimized results showed various effects on the production of isoeugenol glucoside with different conditions such as the production time, substrate concentration, different carbon sources, and media. These experiments were conducted under the previously established conditions of temperature (30 ℃), IPTG condition (0.5 mM), and concentration of the carbon source (10%) by Amoah et al. (2022) [15].

The production time proceeded at 2-h intervals (2, 4, 6, 8, and 10 h) in the experiments. As shown in Figure 4A, the conversion rate was 53% and 64% at 4 and 6 h, respectively, whereas there was an 11% increase in the production of conversion at 6 h than at 4 h (Figure 4A). The substrate concentration (1 mM, 2 mM, 3 mM, 5 mM, and 10 mM) was chosen to clarify the optimum conversion. From the highest conversion rate (64%) with 1 mM, the conversion rate decreased as the substrate concentration increased (Figure 4B).

Furthermore, *C. glutamicum* can obtain carbon from a variety of sources for growth and energy, including monosaccharides such as glucose, fructose, and ribose, disaccharides such as sucrose, mannose, and maltose, alcohols such as inositol or ethanol, organic acids such as pyruvic acid, propionic acid, lactic acid, acetic acid, and gluconic acid, as well as some amino acids such as L-glutamate and L-glutamine [40,41]. Various carbon sources, such as fructose, maltose, and glucose, were chosen as the candidate sources to ascertain the optimal carbon source for the bioconversion of isoeugenol glucoside derivatives (Figure 4C). Isoeugenol glycoside was synthesized, and after 5 h, around 93% of the production rate was shown in the sucrose-fed sample (Figure 4C).

In this study, we evaluated the effect in various media to compare the effectiveness between an enriched medium and a minimal medium, which is widely used for large-scale production in *C. glutamicum*. The CGXII minimal medium is a highly defined medium for *C. glutamicum* employed for both basic and practical research [42]. The process of formulating a medium is typically one that is arduous, expensive, and time-consuming because the ideal media for one strain may not be suitable for another [43]. Additionally, some studies investigated the influence of a BHI medium on the kinetic parameters and growth behavior of *C. glutamicum* ATCC 13032 [44], and the production of scyllo-inositol from myo-inositol in a rich BHI medium [45].

Herein, we examined four enriched media that are commonly used for cultivation, such as LB, LBG, BHI, and BHIS, to compare them with the CGXII minimal medium at 30 ℃ after induction with IPTG concentration (0.5 mM) to find the optimal medium for isoeugenol glucoside production. When the optical density (OD_600_) reached 2.0, the culture medium was extracellularly supplemented with 10% glucose and 2 mM isoeugenol as substrates. In LBG, it showed the highest conversion rate of about 76% among the other media conditions, furthermore, CGXII showed the lowest conversion rate (15%) among others. A possible explanation for the difference is that BHI relieved the strain of the energy-intensive biomass synthesis, most likely of proteins and fatty acids, subsequently reducing the chemical energy source demand of the cell, such as reducing ATP and NADPH, because all amino acids must be produced from glucose and ammonia in the CGXII medium, which is linked to a high NADPH demand [44,45] (Figure 4D).

### 2.4. Water Solubility Test

Products **1b** and **1c** were found in both the ethyl acetate and aqueous layer. Product **1b,** confirmed to be isoeugenol-1-*O*-β-d-glucoside, was predominant in the aqueous phase, whereas a more acetylated compound was present in the organic layer, due to the fact that the acetyl group is hydrophobic. Because it has three more hydroxyl groups (–OH) than isoeugenol, the glycosylated molecule possesses effective hydrogen bonding. The water solubility of glucosylated isoeugenol was approximately 310-fold higher than that of isoeugenol, and 7.7-fold higher than that of the acetylated glucoside. In comparison to the substrate, the glucosylated product had a considerably higher solubility in water (Appendix A).

### 2.5. Modification of Monoterpenes by C. glutamicum pSKSM-YdhE

The effectiveness of the developed system towards the glucosylation of other monoterpenes was tested, thus, eugenol, thymol, and carvacrol were chosen for further studies (Appendix A). These substrates were exogenously supplemented in *C. glutamicum* harboring the pSKSM-YdhE cell culture under 200 rpm of agitation; 2 mM of the substrate and 10% glucose were in BHIS media for 5 h as previously mentioned in the materials and methods. The HPLC-PDA with UV absorbance at 280 nm showed the bioconversion of two eugenol glucoside derivatives with retention times of t_R_ 9.65 min and t_R_ 11.36 min, similar to that of isoeugenol (Appendix A). The HR-QTOF ESI/MS analysis then further confirmed the two products, with a molecular weight of *m*/*z* 349.1272 and *m*/*z* 391.1375 in sodium mode, respectively, which matched with the calculated masses. Surprisingly, only one glucosylated product was obtained when thymol and carvacrol were used as substrates. These products were revealed as thymol and carvacrol glucosides as confirmed by HPLC and HR-QTOF ESI/MS (Appendix A). As a result, the strain showed the ability to convert four of the monoterpene substrates to their respective *O*-linked glucoside derivatives (Table 2, Appendix A).

## 3. Methods and Materials

### 3.1. General Procedures

*Corynebacterium glutamicum* ATCC 13032 (*C. glutamicum*) served as an expression host for the production. Brain heart infusion broth medium (BHI) was used to cultivate *C. glutamicum* (10 g/L tryptone, 5 g/L yeast extract, 10 g/L brain heart infusion broth, and 10 g/L NaCl) (Difco, Franklin Lakes, NJ, USA) at 30 °C in a shaking incubator at 200 rpm. BHIS (BHI medium supplemented with 91 g/L sorbitol) medium was employed together with the necessary antibiotics to generate transformants in *C. glutamicum* 15 g/L of bacteriological agar (Sigma, St. Louis, MO, USA), which was added to make an agar plate. For the production, *C. glutamicum* was used as the expression host. Luria Broth (LB) medium and LB medium with 20 g/L glucose (LBG), as well as CGXII minimal medium [46] were used to define the optimal media for the best conversion rate. CGXII minimal medium contained: 20 g/L (NH_4_)_2_SO_4_, 5 g/L urea, 1 g/L KH_2_PO_4_, 1 g/L K_2_HPO_4_, 0.25 g/L MgSO_4_·7 H_2_O, 42 g/L 3-morpholinopropanesulfonic acid (MOPS), 10 mg/L CaCl_2_, 10 mg/L FeSO_4_·7 H_2_O, 10 mg/L MnSO_4_ · H_2_O, 1 mg/L ZnSO_4_·7 H_2_O, 0.2 mg/L CuSO4, 0.02 mg/L NiCl_2_·6 H_2_O, 0.2 mg/L biotin, 40 g glucose, and 30 mg protocatechuic acid (PCA). The media and the glucose solution were autoclaved separately at 115 °C for 30 min, and the needed amounts of glucose were then added.

### 3.2. Generation of Recombinant Strains

A previously constructed vector, pSKSM-YdhE, harboring an inducible promoter, a ribosome-binding site, and a codon-optimized YdhE [15] was employed for this study. *C. glutamicum* electrocompetent cells were prepared as previously described [15]. Briefly, 50 mL of BHIS medium supplemented with 1 mL/L Tween 80 and 500 μL of seed culture were injected and shaken at 200 rpm until the optical density (OD_600_) reached 0.8. Cells were placed on ice for 20 min to chill them, and they were rinsed four times with chilled 10% glycerol and centrifuged for 10 min at 3500 rpm at 4 °C. Cell pellets were reconstituted in 150 μL aliquots and kept at −80 °C after being resuspended in 1 mL of 10% glycerol. The cells were electroporated, recovered for 2 h at 30 °C with 160 rpm shaking, then plated on BHIS plates containing 50 mg/L kanamycin (km), where they were allowed to develop for 1–2 days. By digesting the plasmid with a restriction enzyme, positive clones were confirmed.

### 3.3. Determination of Optimal Culture Conditions

Engineered *C. glutamicum* carrying the recombinant plasmid pSKSM-YdhE was cultivated in 250 mL sterile flasks to ascertain the ideal culture conditions for the in vivo conversion of CO-YdhE. Cell culture was induced with 0.5 mM IPTG for 12 h at 30 °C when the OD_600_ reached 0.8. The induced cells were fed with different concentrations of isoeugenol (1, 2, 3, 5, and 10 mM) and different carbon sources (fructose, glucose, maltose, and sucrose). Both experiments were studied at 30 °C for 5 h. To determine the optimal medium with BHI, BHIS, LB, LBG, and CGXII, the cells were inoculated and fed in each different medium for 5 h at 30 °C.

Samples (1 mL) were taken at 2 h intervals for the analysis to determine the optimal production time. For removing the cell pellets, the samples were mixed well with an equal volume of cold methanol before being vortexed for 20 min at 4 °C. The supernatant was gathered, filtered, and subjected to HPLC analysis before being subjected to mass spectrometry analysis. All tests were performed in triplets, and *C. glutamicum*, which solely carried pSK003, was used as the control.

### 3.4. In Vivo Glucosylation of Monoterpenes

According to the position of the hydroxyl group, we conducted the in vivo glucosylation of several monoterpenes. In a 5 mL BHI medium with 200 rpm shaking at 30 °C, a single colony of *C. glutamicum* carrying pSKSM-YdhE was grown for up to 16 h. A total of 500 μL of seed culture was inoculated into 50 mL of fresh BHIS medium, and the mixture was stirred continuously until the OD_600_ reached approximately 0.8. The cell culture was then induced with 0.5 mM IPTG. The addition of prepared monoterpenes (2 mM) and sterilized glucose (10%) to the cell culture were exogenously supplied after 10 h. The cells were grown for a further five hours. We collected samples (1 mL) for analysis at various time points. To eliminate cell pellets, the samples were mixed well with an equal volume of cold methanol before being vortexed for 30 min at 4 °C. The supernatant was collected, filtered, and analyzed using HPLC before being examined further using LC/MS spectrometry. All tests were performed in triplets, and the *C. glutamicum* used for the control trials only contained pSK003. Monoterpenes are listed in Table 2 and Appendix A.

### 3.5. Water Solubility Test

The water solubility of isoeugenol and its derivatives was determined as previously reported by Thapa et al., 2019 [47]. In brief, equal volumes of cell culture and ethyl acetate were mixed and vortexed for 20 min. Then, after the mixture was centrifuged for 30 min at 4 °C at 13,000 rpm, both the aqueous and ethyl acetate layers were separated, collected, and evaluated directly with HPLC-PDA. The substrate and two products, isoeugenol glucoside and acetylated glucoside were evaluated directly by HPLC-PDA in both the water and solvent (ethyl acetate) fractions.

### 3.6. Fermentation and Analytical Procedures

The fermentation procedure was carried out using *C. glutamicum* pSKSM-YdhE. Recombinant bacteria were grown in 5 L flasks at 30 °C in BHIS medium with additional thiamin and Tween 80 at 0.001% (*v*/*w*) and 0.1% (*v*/*w*), respectively. After induction with 0.5 mM IPTG, the fermentation was allowed to continue for 6 h with the addition of 1 mM isoeugenol and 10% glucose (*v*/*v*). Double volumes of ethyl acetate were used to extract the products, and they were then evaluated using high-performance liquid chromatography with a photodiode array (HPLC-PDA) that could be seen at 255 nm. In the mobile phase, acetonitrile (ACN) and water (with 0.1% trifluoroacetic acid) were utilized. The ACN concentrations were as follows: 0%, (0–6 min); 50%, (6–10 min); 100%, (10–14 min); 100%, (14–20 min); 0%, (20–22 min), with a flow rate of 1 mL/min.

To confirm the glucosylated products, the samples were analyzed using an ultraperformance liquid chromatography–mass spectrometry (UPLC-MS) system. An Acquity UPLC unit (Waters, Milford, MA, USA) was equipped with an Acquity UPLC BEH C18 column (2.1 mm × 100 mm, 1.7 μm). The column was eluted at a flow rate of 0.3 mL/min at 45 °C with gradients in mobile phase A (0.1% (*v*/*v*) trifluoroacetic acid) and B (100% (*v*/*v*) acetonitrile). The mobile phase was changed from 5% A/95% B to 90% A/10% B for 12 min. The eluent from the column was directed to a WATER SYNAPT G2-S-Q-TOF MS instrument (Milford, MA, USA) working on electrospray ionization (ESI) positive mode using the following parameters: capillary, 2.5 kV; cone, 30 V; source block temperature, 120 °C; desolvation temperature, 300 °C; desolvation gas flow: 600 L/h; cone gas flow: 10 L/h; collision energy: 40 eV; mass range (*m*/*z*): 20–1000.

The compounds were purified using a preparative ultimate 3000 UPLC (Thermo Fisher Scientific, Waltham, MA, USA) with a C_18_ column (YMC-Pack ODS-AQ (250 mm × 20 mm I.D, 10 μm Shimogyo-ku, Kyoto, Japan) connected to a UV detector at 255 and 360 nm over the course of a 20 min binary program with 100% triple-distilled water and ACN (10%, (0–6 min); 50%, (6–10 min); 100%, (10–14 min); 100%, (14–20 min); 0, (20–22 min)) at a flow rate of 10 mL/min.

A rotary evaporator was used to purify and concentrate the products before they were dissolved in deuterium oxide and lyophilized. The completely dried samples were freeze-dried and then dissolved in 600 μL of dimethyl sulfoxide-*d*_6_ (DMSO-*d*_6_) for nuclear magnetic resonance (NMR). The NMR analysis of the isoeugenol and its glycosylated derivative was performed. By employing a Cryogenic TCi probe and a 700 MHz Avance II 900 Bruker BioSpin NMR spectrometry (Bruker, Billerica, Massachusetts, USA), the purified chemicals (5 mm) were analyzed. Furthermore, one-dimensional NMR (^1^H NMR, ^13^C NMR) and two-dimensional NMR (COSY) were used to clarify the compounds’ structures. The Topspin 3.1 program (Bruker, Billerica, MA, USA) was used to handle all raw data. MestReNova 12.0 software (Mestrelab Research S.L. Santiago de Compostela, A Coruña, Spain) was then used to perform an additional analysis.

### 3.7. Statistical Analysis

The bioconversion rate data were analyzed using a one-way analysis of variance (ANOVA), followed by Dunnett’s multiple comparison test using GraphPad Prism version 9 for Windows (GraphPad Software, San Diego, CA, USA, www.graphpad.com, accessed on 15 April 2023) and the Excel statistical application (Microsoft 365, Microsoft Corporation, Redmond, WA, USA).

## 4. Conclusions

This study introduces the successful in vivo production of derivatives of monoterpenes, isoeugenol, eugenol, thymol, and carvacrol with *C. glutamicum* expressing glycosyltransferase, YdhE. The bioconversion of isoeugenol showed two products, which, through NMR analysis, were revealed to be isoeugenol-1-*O*-β-d-glucoside (**1b**) and isoeugenol-1-*O*-β-d-(2″-acetyl)-glucoside (**1c**), which are the modifications of the glucoside compound by the native sugar-*O*-acetyltransferase present in *C. glutamicum,* and they may also exhibit improved properties. For instance, Hao et al., succeeded in the acetylation of anthocyanin by the enzymatic acylation of pelargonidin-3-*O*-glucoside by *Candida antarctica* lipase B, which showed that acetylated pelargonidin-3-*O*-glucoside has promising thermostability and lipophilicity, as well as being protective against oxidative damage via the activation of the Nrf2/ARE pathway [48].

Additionally, a number of factors, including the production time, substrate concentration, carbon source, and culture medium, were optimized to ensure the maximum biosynthesis of these products. The results showed a maximum production within six hours after the extracellular supplementation of 1 mM of the substrate and 10% sucrose in the LBG medium. Interestingly, the water solubility determination showed a 310-fold higher solubility of the glucosylated compound than isoeugenol.

Overall, our research introduces the biosynthesis of isoeugenol glucoside and acetylated isoeugenol glucoside in engineered *C. glutamicum*, which can be used as a potential host for the industrial-scale production and glucosylation of therapeutic *O*-linked compounds, highlighting the feasibility of using glucose as a source of glycol diversifying natural and non-natural products, including *O*-linked compounds, in a highly sustainable and cost-effective system. In addition, the post-translational modifications of monoterpenes in *C. glutamicum* with the in vivo system with a high production yield provides a feasible industrial-scale biosynthesis of modified compounds with improved solubility, half-life, immunogenicity, and stability.

## Figures and Tables

**Figure 1 molecules-28-03789-f001:**
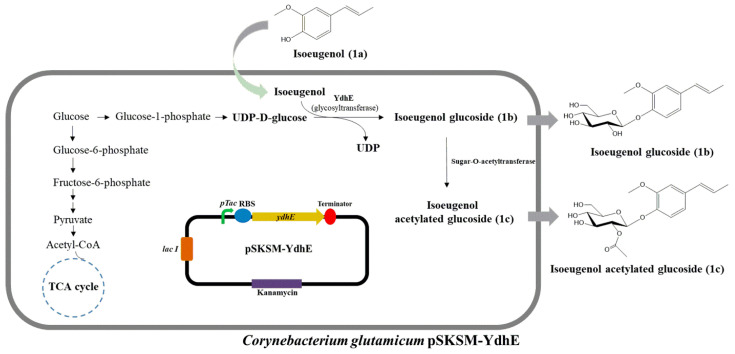
Glucosylation of isoeugenol by *Corynebacterium glutamicum* pSKSM-YdhE. Sugar *O*-acetyltransferase attaches the acetyl group to the glucosylated compound.

**Figure 2 molecules-28-03789-f002:**
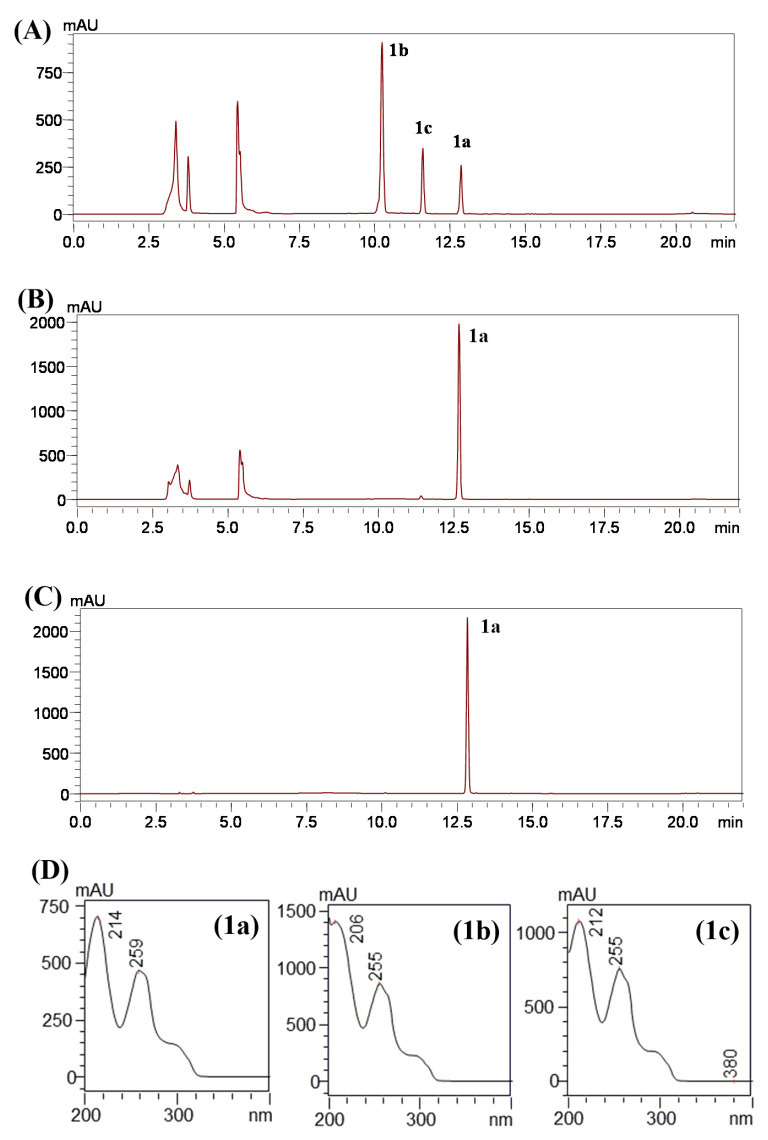
In vivo bioconversion of isoeugenol (**1a**) to isoeugenol-1-*O*-β-d-glucoside (**1b**) and isoeugenol-1-*O*-β-d-(2″-acetyl)-glucoside (**1c**) in *C. glutamicum* pSKSM-YdhE. (**A**) HPLC chromatogram analysis of whole-cell biotransformation with CO-YdhE in *C. glutamicum*, (**B**) whole-cell biotransformation with pSK003 vector in *C. glutamicum.* (**C**) Isoeugenol standard (1 mM), (**D**) UV spectra of isoeugenol (**1a**), and glucoside derivatives **1b** and **1c**.

**Figure 3 molecules-28-03789-f003:**
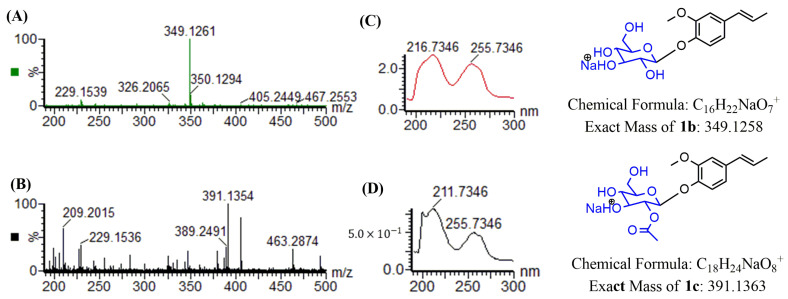
HR-QTOF ESI/MS analysis. (**A**) Isoeugenol-1-*O*-β-d-glucoside (**1b**), (**B**) Isoeugenol-1-*O*-β-d-(2″-acetyl)-glucoside (**1c**), (**C**) UV/VIS of (**1b**), (**D**) UV/VIS of (**1c**).

**Figure 4 molecules-28-03789-f004:**
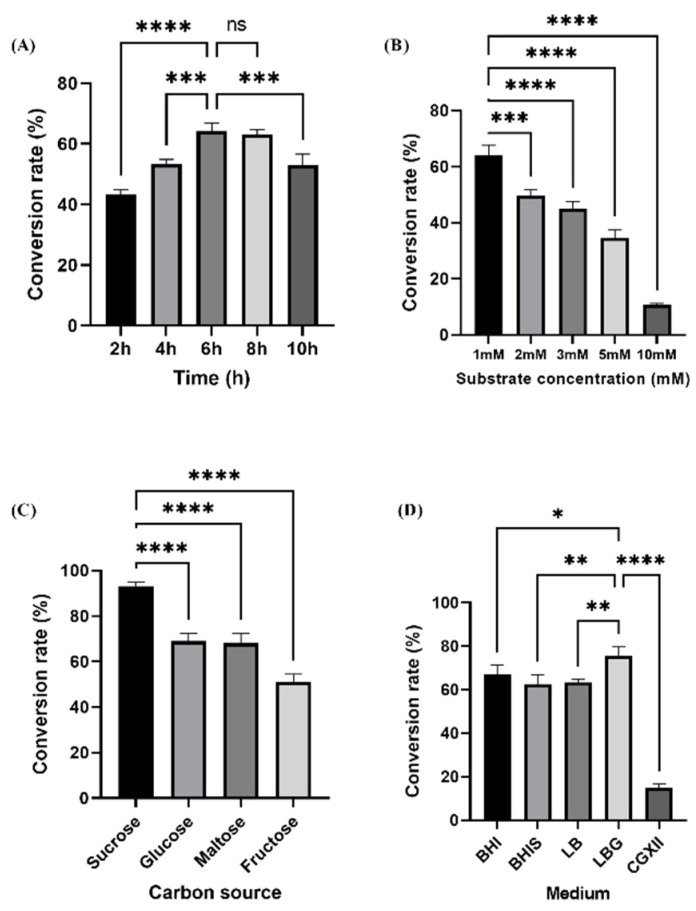
Determination of optimum culture conditions using isoeugenol as substrate. (**A**) Production time length at a 2-h time interval (*n* = 5). (**B**) The conversion rate of isoeugenol glucoside with different substrate concentrations (*n* = 5). (**C**) The conversion rate of isoeugenol with different carbon sources (*n* = 4). (**D**) The conversion rate in different media (*n* = 5). Error bars show the standard deviation of three distinct experiments. One-way ANOVA was used for the statistical analysis. Significant differences among means are indicated by asterisks in each graph (ns indicates non-significant, * indicates *p* < 0.05, ** indicates *p* < 0.01, *** indicates *p* < 0.001, **** indicates *p* < 0.0001).

**Table 1 molecules-28-03789-t001:** ^1^H NMR and ^13^C NMR of isoeugenol-1-*O*-β-d-glucoside (**1b**) and isoeugenol-1-*O*-β-d-(2″-acetyl) glucoside (**1c**).

No	Isoeugenol-1-*O*-β-d-glucoside (1b) (DMSO-*d*_6_)		Isoeugenol-1-*O*-β-d-(2″-acetyl)-glucoside (1c) (DMSO-*d*_6_)	
	^13^C	^1^H (Multiplicity, *J*)	^13^C	^1^H (Multiplicity, *J*)
1′	100.50	4.87 (1H, d, *J* = 7.2 H)	100.27	4.91 (1H, d, *J* = 7.0 Hz)
2′	73.67	3.31–3.24 (1H, m)	73.60	3.28 (1H, q, *J* = 7.1 Hz)
3′	77.29	3.31–3.24 (1H, m)	74.04	3.28 (1H, q, *J* = 7.1 Hz)
4′	70.12	3.31–3.24 (1H, m)	70.29	3.18 (1H, m)
5′	77.45	3.16 (1H, t, *J* = 8.9 Hz)	77.01	3.57 (1H, ddd)
6′	61.12	3.45 (1H, d, *J* = 12.3 Hz)	63.84	4.07 (1H, dd, *J* = 11.9, 6.8 Hz)
		3.67 (1H, d, *J* = 11.5 Hz)		4.25 (1H, dd, *J* = 11.9, 2.1 Hz)
1″			170.70	
2″			21.12	1.99 (3H,s)
1	132.07		132.26	
2	130.99		130.96	
3	118.89	7.00 (1H, d, *J* = 8.4 Hz)	118.75	7.02 (1H, d, *J* = 2.0 Hz)
4	124.16		124.30	
5	109.98	6.84 (1H, d, *J* = 2.1Hz)	110.02	6.96 (1H,d, *J* = 2.1Hz)
6	115.74	7.00 (1H, d, *J* = 8.4 Hz)	115.75	6.85 (1H, dd, *J* = 8.4, 2.0 Hz)
7	149.44	6.44 (1H, dd, *J* = 16.1, 1.9 Hz)	149.46	6.34 (1H, dd, *J* = 15.9, 2.0 Hz)
8	146.10	6.19 (1H, dq, *J* = 15.9, 6.6 Hz)	145.83	6.19 (1H, dq, *J* = 15.8, 6.6 Hz)
9	18.64	1.82 (3H, d, *J* = 6.5 Hz)	18.63	1.82 (3H, d, *J* = 6.5Hz)
10	56.06	3.77 (3H, s)	56.05	3.77 (3H, s)

**Table 2 molecules-28-03789-t002:** Bioconversion of monoterpene glucoside compounds after 5 h of incubation.

Name	Conversion Rate (%)		Total (%)
	Glucoside	Acetylated Glucoside	
Isoeugenol	58.5%	24%	82.5%
Eugenol	25%	12%	37%
Thymol	52%	-	52%
Carvacrol	27%	-	27%

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
