# Peer review of "Glucosylation of Isoeugenol and Monoterpenes in Corynebacterium glutamicum by YdhE from Bacillus lichenformis"

_molecules, 2023, doi:10.3390/molecules28093789_

Round 1

Reviewer 1 Report

The authors developed a methodology for the production of glycosyl derivatives of isoeugenol and other monoterpenes using biotransformation. The manuscript has some issues to be considered before its publication in Molecules. Please find below some aspects to be improved before its acceptance.

(1) the manuscript title contains only the “isoeugenol” word and not the name of other investigated compounds. The abstract is little informative regarding this aspect and the optimal conditions identified.

(2) Figure 1 must be better explained. Some of the aspects depicted by the figure are absent in the manuscript text.

(3) found mass for compound 1c showed 102 ppm of error in relation to the theoretical mass (m/z = 391.13 and m/z = 391.17). This is a value unacceptable in the chemical identification of the analyte. Please check this aspect in all the manuscript for others compounds, mainly in compounds not characterized with NMR. A poor agreement in the mass value is not suitable for the identification of the analyte.

(4) Only two decimal cases were used in the m/z values in the manuscript text, however, in the HRMS spectra figures were obtained with four decimal cases.

(5) please check the purity degree of the compound used for the spectra of figure S1 obtaining. The alkene signals near 170 ppm was not assigned in figure S1.

(6) figure captions in electronic supporting information must have the solvent used for spectra acquisition and the NMR frequency for each nucleus.

(7) the font size in the NMR spectrum is small. It's difficult to see the chemical shift values of the signals.

(8) the sentence “Water solubility test performed as mentioned in materials and methods” is not informative and unnecessary.

(9) Methodology not reported the spectral description. Please see the authors instructions on the Molecules site.

(10) is missing statistical analysis in the graphs of figure 4. is strongly suggested the use of variance analysis, aiming for an improved description of the variables effect on conversion rate. The figure caption contains information p < 0.05, but no statistical analysis was performed. See that in the manuscript is written “conversion rate” and in the graphs depicted in Figure 4 are written “concentration”.

(11) the column “intensities” in table 1 is confused

(12) it is unclear whether the catalytic activity of glucosyl transferase was previously investigated

(13) conclusion is too long and is not limited to highlighting the main findings of the authors. Were cited many references in the manuscript conclusion. Please check the conclusion style.

Author Response

The authors developed a methodology for the production of glycosyl derivatives of isoeugenol and other monoterpenes using biotransformation. The manuscript has some issues to be considered before its publication in Molecules. Please find below some aspects to be improved before its acceptance.

Answer; We are very thankful to the reviewer for their kind suggestion aimed at for improving the manuscript. According to the reviewer’s suggestion, we have changed the focus of the manuscript and thoroughly revised the manuscript

(1) the manuscript title contains only the “isoeugenol” word and not the name of other investigated compounds. The abstract is little informative regarding this aspect and the optimal conditions identified.

Answer; The manuscript title has been appropriately modified as indicated in the manuscript title.

(2) Figure 1 must be better explained. Some of the aspects depicted by the figure are absent in the manuscript text.

Answer; The appropriate modifications and explanations have been added to the introduction section of the manuscript.

(3) found mass for compound 1c showed 102 ppm of error in relation to the theoretical mass (m/z = 391.13 and m/z = 391.17). This is a value unacceptable in the chemical identification of the analyte. Please check this aspect in all the manuscript for others compounds, mainly in compounds not characterized with NMR. A poor agreement in the mass value is not suitable for the identification of the analyte.

Answer; We are thankful to reviewer for pointing out an obvious typing mistake on our part. Appropriately, the calculated mass have been duly corrected in the manuscript.

(4) Only two decimal cases were used in the m/z values in the manuscript text, however, in the HRMS spectra figures were obtained with four decimal cases.

Answer; We have appropriately corrected the m/z values in the manuscript to four decimal cases as reviewer suggested.

(5) Please check the purity degree of the compound used for the spectra of figure S1 obtaining. The alkene signals near 170 ppm was not assigned in figure S1.

Answer; The alkene signal near 170 ppm is an impurity as a result of the in vivo bioconversions. This impurity however did not interfere with the structural determination of the structure of the compounds obtained.

(6) figure captions in electronic supporting information must have the solvent used for spectra acquisition and the NMR frequency for each nucleus.

Answer; The appropriate information has been duly added to the electronic supporting file.

(7) the font size in the NMR spectrum is small. It's difficult to see the chemical shift values of the signals.

Answer; The font size in the NMR spectrum has been duly increase to enhance visibility as reviewer suggested

(8) the sentence “Water solubility test performed as mentioned in materials and methods” is not informative and unnecessary.

Answer; The appropriate information has been duly corrected in the manuscript.

(9) Methodology not reported the spectral description. Please see the authors instructions on the Molecules site.

Answer; The appropriate information has been duly added to the manuscript.

(10) is missing statistical analysis in the graphs of figure 4. is strongly suggested the use of variance analysis, aiming for an improved description of the variables effect on conversion rate. The figure caption contains information p < 0.05, but no statistical analysis was performed. See that in the manuscript is written “conversion rate” and in the graphs depicted in Figure 4 are written “concentration”.

Answer; As reviewer suggested, we have accordingly revised these graphs by using One-way ANOVA, because our figures contained 4 or above of independent variables and one dependent variable so that we choose to use one-way ANOVA. We also add more information of statistical analysis such as column number of each results with (n), significant differences of among means by asterisks (*), and also changed title of y axis as ‘conversion rate’.

(11) the column “intensities” in table 1 is confused

Answer; ; The appropriate information has been duly corrected in Table 1.

(12) it is unclear whether the catalytic activity of glucosyl transferase was previously investigated

Answer; Our group had previously investigated the catatylic ability of glycosyltransferases from Bacillus lichenformis including YdhE in E. coli (Panday et.al. 2019). The current research however, focuses on the application of the said glycosyltransfrase towards the large scale production of isoeugenol and other monoterpenes.

 (13) conclusion is too long and is not limited to highlighting the main findings of the authors. Were cited many references in the manuscript conclusion. Please check the conclusion style.

Answer; As reviewer suggested, conclusion has been duly revised.

Reviewer 2 Report

The authors have demonstrated glucosylation of isoeugenol in C. glutamicum. The study is interesting and are one of the few reports on glucosylation in C. glutamicum. A few things I would like to point out for authors to consider.

a) My main concern is regarding the novelty of the study. The authors have published a similar study last year in the same journal. On a closer look, both studies appear more and less similar with a few differences including the targeted molecule for glucosylation. Moreover experimental design is also quite similar probably it is due to the reason that same constructs were used.

b) I do not understand why all the experiments were performed in enrich media (LB, BHI, LBG etc). CgXII media is the well established media for performing production purposes in C. glutamicum. Authors have tested multiple sugars (glucose, fructose, maltose, sucrose) so minimla media seems apt for such experiments instead of enrich media.

c) Introduction and conclusion can be improved and recent literature need to be discussed. Current introduction does not discuss the benefit of using C. glutamicum over other chassis such as widely employed E. coli. In current form the manuscript looks mirror of author's previous publication. I will recommend authors to discuss some of the relevant publications to make this comparison. A few suggestions for authors to consider doi.org/10.1016/j.crbiot.2021.12.004, doi:10.3389/fbioe.2021.748510, doi.org/10.1038/s41589-020-0595-9doi.org/10.1186/s12934-018-0990-z, doi: 10.1186/s12934-018-1013-9

Author Response

The authors have demonstrated glucosylation of isoeugenol in C. glutamicum. The study is interesting and are one of the few reports on glucosylation in C. glutamicum. A few things I would like to point out for authors to consider.

Answer; We are very much grateful to reviewer for the kind words and suggestions aimed at improving and enriching the content of our manuscript.

a) My main concern is regarding the novelty of the study. The authors have published a similar study last year in the same journal. On a closer look, both studies appear more and less similar with a few differences including the targeted molecule for glucosylation. Moreover experimental design is also quite similar probably it is due to the reason that same constructs were used.

Answer; Our previous studies focused on the the developed system’s ability to glucosylate therapeutic N-linked compounds on Corynebacterium glutamicum. However, the current research focuses on the flexibility and the application of the system towards isoeugenol and other O-linked monoterpenes.

b) I do not understand why all the experiments were performed in enrich media (LB, BHI, LBG etc). CgXII media is the well established media for performing production purposes in C. glutamicum. Authors have tested multiple sugars (glucose, fructose, maltose, sucrose) so minimla media seems apt for such experiments instead of enrich media.

Answer; As reviewer noted, CGXII and its related minimal media has been the established media for production purposes in C. glutamicum. However, in our current research, we aim to be able to biosynthesize glucosylated products on a large scale with minimal cost and labor. Furthermore Ramp et. el., 2021 reported high production of Scyllo-inositol in C. glutamicum with BHI media compared to CGXII, thus BHI as a rich medium generally used for growth has also proven to be an avid candidate for production purposes in C. glutamicum.

c) Introduction and conclusion can be improved and recent literature need to be discussed. Current introduction does not discuss the benefit of using C. glutamicum over other chassis such as widely employed E. coli. In current form the manuscript looks mirror of author's previous publication. I will recommend authors to discuss some of the relevant publications to make this comparison. A few suggestions for authors to consider org/10.1016/j.crbiot.2021.12.004, doi:10.3389/fbioe.2021.748510, doi.org/10.1038/s41589-020-0595-9, doi.org/10.1186/s12934-018-0990-z, doi: 10.1186/s12934-018-1013-9

Answer; As reviewer suggested, introduction part has duly been with respect to the current and relevant publications

Round 2

Reviewer 1 Report

The manuscript was improved in many aspects and is currently suitable for publication in Molecules. I have only two minor considerations in the current step of the editorial process:

(1) in figure 4, the horizontal bars have inaccurate limits. It is unclear what bars are different from one another.

(2) conclusion is too long and is not limited to highlighting the main findings of the authors. Were cited many references in the manuscript conclusion. Please check the conclusion style.

Author Response

Comments;

The manuscript was improved in many aspects and is currently suitable for publication in Molecules. I have only two minor considerations in the current step of the editorial process:

Answer; We are grateful to the reviewer for the kind words and suggestion aimed at for improving the manuscript. Accordingly, reviewer’s consideration has been kindly addressed in the manuscript.

(1) in figure 4, the horizontal bars have inaccurate limits. It is unclear what bars are different from one another.

Answer; We have duly corrected with accurate horizontal lines to show significant differences between each bar with highest mean in each figure.

(2) conclusion is too long and is not limited to highlighting the main findings of the authors. Were cited many references in the manuscript conclusion. Please check the conclusion style.

Answer; According to reviewer’s suggestion, we have duly modified the conclusion part of the manuscript, keeping fewer references as possible and highlighting our main finding.

Reviewer 2 Report

The authors have provided point to point response to the concerns raised by me in my previous review.  I am satisfied with modified version. Therefore, I endorse the manuscript for publication.

Author Response

The authors have provided point to point response to the concerns raised by me in my previous review. I am satisfied with modified version. Therefore, I endorse the manuscript for publication

Answer; We are very thankful to reviewer and very much appreciate the reviewer’s effort to improve our manuscript.